# Individual and Social Level Factors Influencing Repeated Pregnancy among Unmarried Adolescent Mothers in Katavi Region—Tanzania: A Qualitative Study

**DOI:** 10.3390/children9101523

**Published:** 2022-10-05

**Authors:** Salim Juma Mpimbi, Mwajuma Mmbaga, Ziad El-Khatib, Minyahil Tadesse Boltena, Samwel Marco Tukay

**Affiliations:** 1Monitoring Evaluation and Research Expert/Technical Advisor, ASK Aria Consulting, Dar es Salaam 16114, Tanzania; 2Department of Nursing and Midwifery, Katavi Referral Regional Hospital, Katavi P.O. Box 449, Tanzania; 3Department of Global Public Health, Karolinska Institutet, 17176 Stockholm, Sweden; 4World Health Programme, Université du Québec en Abitibi-Témiscamingue (UQAT), Rouyn-Noranda, QC J9X 5E4, Canada; 5Armauer Hansen Research Institute, Ministry of Health, Addis Ababa 1005, Ethiopia; 6Ethiopian Evidence Based Health Care Centre: A JBI Center of Excellence, Public Health Faculty, Institute of Health, Jimma University, Jimma P.O. Box 378, Ethiopia; 7The Nelson Mandela African Institution of Science and Technology, Arusha P.O. Box 447, Tanzania; 8Pangani District Hospital, Pangani P.O. Box 89, Tanzania; 9Ifakara Health Institute (IHI), Dar es Salaam P.O. Box 78373, Tanzania

**Keywords:** adolescents, individual, repeated pregnancy, social, Tanzania, unmarried, Katavi

## Abstract

Adolescents’ pregnancy rates are still high in Tanzania, despite the efforts made by the national campaign. Within two years after the first pregnancy, adolescent mothers are more at risk of repeat conception. Repeated pregnancies are associated with increased maternal and perinatal outcomes. Katavi is a leading region in the country, with 45% adolescent pregnancy. Studies are scarce on factors influencing repeated pregnancy among unmarried adolescent mothers in the region. Therefore, this study explored the individual and social level factors influencing repeated pregnancy among unmarried adolescent mothers in the Katavi Region. An exploratory qualitative study, using key informant interviews (KIIs) was adopted for 16 participants. The study participants were unmarried adolescent mothers, aged 15–19 years, who were purposively sampled. Thematic analysis was used to analyze qualitative data. QSR Nvivo version 14 was used to analyze these data. The study established the individual factors influencing repeated pregnancy, which were inadequate sexuality knowledge, individually perceived barriers to contraceptive use, and the guarantee for marriage. Furthermore, the social factors identified were the power of decision-making, peer pressure, and the parent–child relationship. Inadequate education on sexuality is observed as a crucial factor influencing repeated pregnancy. Parents as primary educators should be encouraged to talk with their children, especially adolescent girls about sexual education.

## 1. Introduction

Adolescence refers to the years of transition from childhood to adulthood, between 10–19 years of age. It is a stage of significant growth, increasing independence, vulnerability, experience, and major physical and psychological changes. Physical, emotional, and cognitive maturation influence how individuals experience their adolescent years. Almost 20% of the world’s population is made of teenagers [1].

Globally, approximately 16 million young women aged 15 to 19, which is about 11% of all births worldwide, and about 1 million girls under 15 years of age give birth every year. Ninety-five percent of these births occur in low- and middle-income countries (LMICs), almost half of which (49%) are unintended [2]. This result is nearly 16 million deliveries, and over 3.2 million had delivery below 28 weeks of gestation age yearly. Although the majority of adolescent births occur in LMICs, there are substantial regional differences, with the highest adolescent birth rates in West and Central Africa and lowest in East Asia. In Sub–Saharan Africa, about 20–40% of teenage girls have been found to become pregnant by the age of 18 [3].

Several factors influencing unintended pregnancies amongst adolescents have been reported; these include early marriages, culture, religion, gender, poor social and economic support, peer pressure, lack of comprehensive sexual education, poor reproductive health services provision, and poor attitude of health workers in providing contraceptive services for adolescents [4]. Additionally, the unmet need for contraceptives by adolescents and fear of contraceptive side effects seem to be a barrier to contraceptive use [4].

Globally, Tanzania is one of the countries with the highest rate of adolescent pregnancy. It is estimated that 23% of girls aged 15–19 years begin childbearing, and 39% of adolescent girls by 18 years old are either already mothers or pregnant. Early childbearing places girls’ health at risk as adolescents in the 15 to 19 years old age group are twice as likely to die in childbirth, as well as being prone to seek unsafe abortion procedures, which have caused death and disability in countries where abortions are illegal, including Tanzania [1].

Repeated adolescent pregnancy is a major reproductive health concern in Tanzania and shows increasing maternal and perinatal poor outcomes [1]. Adolescent (teenage) mothers have an elevated risk of repeated pregnancy (RP) within two years of their first pregnancy. Considering the impact of teenage pregnancy and childbirth on maternal deaths, as well as the debilitating effects on neonatal and child health outcomes, especially in LMICs, RP leads to a higher risk of preterm births, mental health issues, and developmental problems among children [5].

Under the National Family Planning Costed Implementation Plan (NFPCIP) of 2019–2023, one of the strategic priorities is increasing age-appropriate information about, access to, and use of contraceptives among adolescents and youth aged 10–24 [6]. This priority is, in part, dependent on another strategic priority focused on addressing social norms that hinder individuals from using contraception to space or limit births [6].

Tanzania aims to improve and adopt policies that facilitate adolescent and youth access to contraceptive information and services. Stakeholders identified two opportunities to reach in-school youth—first, by reviewing and rolling out an evidence-based national comprehensive sexual education curriculum to ensure that the content on contraception is strong and evidence-based; second, by revising the National School Health Programme guidelines and strategy to include family planning information [6]. On 15 September 2017, Tanzania’s Katavi region announced a new regional action plan to reduce teenage pregnancy. The plan aimed to reduce the rate to 20% by 2020.

Whilst there has been decreased rate of adolescent pregnancy, over the past few decades, the rate is inadmissibly high. This is despite the efforts to reduce adolescent pregnancy in Tanzania [7]. Katavi was reported to have the highest rate of adolescent pregnancy among adolescents aged 15–19 years in the country (45%) [8]. Katavi has also been reported to have the lowest modern contraceptive prevalence rate of 18% in the country [9].

Several research works have been exploring and identifying factors associated with adolescents’ pregnancy [10,11,12]; however, there is little documentation of the factors influencing repeated pregnancy among unmarried adolescent mothers. Additionally, to the best of our literature search, there is no documented research in Tanzania trying to explore factors influencing repeated pregnancy among unmarried adolescent mothers. Therefore, this study sought to explore individual and social level factors influencing repeated pregnancy among unmarried adolescent mothers in Katavi Region.

### Operational Definitions

Adolescence: According to the World Health Organization (WHO), adolescence refers to the years of transition from childhood to adulthood, between 10–19 years of age [13]. However, this study intended to interview adolescents aged 15–19 years, since adolescent/teenage pregnancy is when a girl aged 15–19 is pregnant with her first child or gives birth [14].

Repeated pregnancy: The incidence of two or more pregnancies before the age of 20 years. According to Balassone, the earlier the first pregnancy, the more likely a second pregnancy will occur—in 80% of the cases, an unintended one [15].

## 2. Methods

### 2.1. Study Context

The study was conducted at Mamba health center in Mlele district, Mpimbwe Municipality, within Katavi Region. It involved unmarried adolescent mothers, aged 15–19 years, who were attending antenatal (ANC) or postnatal care (PNC) at Mamba health center and had more than one pregnancy or more than one child. Katavi region was selected because of the highest prevalence of adolescent pregnancy in the country. The rate of maternal deaths in the region could be higher, since many cases are not reported to relevant authorities [16].

Total population in the Mlele district was 282,568 people, with the population of female at approximately 142,588, and the population of those of reproductive age (15–19 years) was 124,270 [17]. The health center serves people from one district, which has 33 villages and receives 120,000 patients annually, with 10,000 patients monthly. The total delivery of pregnant mothers is 157 per month. The center has outpatient, inpatient, reproductive and child health, operating theater department pharmacy, and laboratory departments, with a capacity of 20 beds in the antenatal and post-natal wards, as well as a capacity of 24 beds for the rest of the department [Unpublished].

### 2.2. Study Design and Sampling Method

The study employed an exploratory design, adopting a narrative qualitative approach. Purposive sampling was used to select 16 unmarried adolescents aged 15–19 years, as they were best to inform the research question and enhance understanding of the phenomenon under study. The study intended to answer the following research question: what are the individual and social level factors that influence repeated pregnancy among unmarried adolescent mothers in Katavi Region?

### 2.3. Data Collection

Data were collected through KIIs. The key informant interview guide for unmarried adolescent mothers was used to guide the researcher in the process of data collection. The discussions were moderated by the researcher in Kiswahili with an open-ended question and recorded by audio recorder with permission from participants. A research assistant who had exposure to qualitative research was trained on how to conduct this study and oriented to adherence to ethical principles. He was responsible for arranging a comfortable and quiet place for the interviews and prepared all items required for the session: audio recorder, interview guide, participant identification number, notebooks, consent forms, and pens. The research assistant was also responsible for notetaking during the interview session to highlight particular interests. He was trained before the commencement of data collection.

All the interviews included the use of an audio recorder, with permission from the participants. Those who accepted to participate were asked to sign the consent form. The principle of saturation was applied, where there are no new data, themes, and coding. Moreover, interview questions were structured to facilitate asking multiple participants the same questions; otherwise, one would not be able to achieve data saturation, as it would be a constantly moving target [18].

### 2.4. Data Analysis

The audio recorded from KIIs were transcribed verbatim and then translated from Kiswahili to English. Thematic analysis was used to analyze the data collected. It is the process of identifying patterns or themes within qualitative data [19]. The analysis was performed as follows. The transcripts were read and re-read to identify the sense of the whole interview. Initial codes were generated. Each segment of data that was relevant to or captured something interesting about the research question was coded. Codes were examined to identify some that fit together into a theme. Furthermore, themes were reviewed and modified, and preliminary themes were developed. Finally, themes were defined. This aimed to identify the essence of what each theme is about (Table 1).

### 2.5. Ethics

Ethical approval from the Muhimbili University of Health and Allied Sciences (MUHAS), Research Ethical Committee (REC), was granted for this study on 13 May 2021, with approval number DA.282/298/01.C/. Permission to carry out the study was sought from the office of the Regional Administrative Secretary, as well as the District Medical Officer. All participants were well-informed about the purpose of the study, and informed consent to participate in the study was obtained from all the participants before being interviewed. For those participants who were under 18 of age, the researcher provided an assent form to their parents or guardians. The objectives of the study were explained to the participants. Participation was voluntary. Participants who were aged 18+ years and willing to participate signed the informed consent, and the assent form was provided to parents/guardians to allow the teenage mothers under 18 to participate. To ensure confidentiality, teenage mothers were interviewed alone, without the presence of guardians or parents. Age, gravidity, and parity were needed, and no names were recorded.

## 3. Results

### 3.1. Socio-Demographic Characteristics

The study involved 16 unmarried adolescent mothers, aged 17–19 years, as key informants. Five of them had a primary level of education, one never went to school, and the rest dropped out while still in primary school. All of them are just staying at home (Table 2).

### 3.2. Individual and Social Level Factors Influencing Repeated Pregnancy among Unmarried Adolescents

Six themes emerged, following analysis of the KIIs data; inadequate sexuality knowledge, individual perceived barriers to contraceptive use, guarantee for marriage (individual factors), decision making, peer pressure, and parent–child relationship (social factors) (Table 3).

#### 3.2.1. Inadequate Sexuality Knowledge

Sexuality knowledge is crucial for preventing unwanted or unplanned pregnancies. The study findings revealed a lack of sexual knowledge associated with a lack of sexual education, poor knowledge of family planning methods, for example, how to use female condoms, and understanding of the menstrual cycle influenced repeated pregnancy among unmarried adolescents. One of the key informants reported that:

“I did not expect to hold pregnancy the second time. It was just bad luck because I did not plan for it. I did not know that I was in my dangerous days. I thought I would not be able to conceive in those days. Unfortunately, it happened.”(KII 01)

Another participant reported that:

“I do not know how condoms can be able to avoid pregnancy and by that time I even did not know in which ways it can prevent pregnancy. Although I was taught still I did not know how it worked.”(KII 04)

#### 3.2.2. Individual Perceived Barriers to Contraceptive Use

The study has identified the perceived barriers to contraceptive use, as characterized by fears of side effects and lacking proper knowledge, regarding contraceptives. Some of the participants reported that contraceptives disturbed their menstrual cycle. Some participants also reported that they never believed that contraceptives could prevent pregnancy. No use of contraceptives leads them to repeated pregnancy. One of the key informants pointed out that:

“I was very ill such that I was bleeding too much. This led me to have low blood pressure (hypotensive). The bleeding happened for the whole year, so I decided to remove the implant.”(KII 07)

#### 3.2.3. Regarding Knowledge of Contraceptives, One of the Participants Reported That:

“I will start making the follow-up now. I never took it seriously earlier. I never believed that they can prevent one from getting pregnant. This is because I never have had received education about family planning and the use of contraceptive methods.”(KII 09)

#### 3.2.4. Guarantee for Marriage

The issue of marriage was posed by key informants as one of the factors influencing repeated pregnancy among unmarried adolescents. Some adolescents reported that they were promised to be married if they accepted having sex with their partners; however, the promises were never fulfilled. Some participants reported that, after being impregnated for the second time, men denied the pregnancy. Others thought carrying a man’s baby (pregnancy) is a guarantee for marriage. It was pointed out by one of the key informants that:

“The first one was due to the girl’s teenage stage but the second one was that the man came looking for the right person to get married to. He ended up luring me that he will marry me. We met one day in the video hall at night and it was the day that we also engaged in sex. After impregnating me, he left the place that he had rented and returned to their home.”(KII 11)

Another participant reported that;

“That is where I was heading because I believed that he was going to marry me because of that I ended up accepting making love with him. When I got pregnant, he rejected it and said that I was brainless. How could I be so stupid to be put in a trap of love?”(KII 05)

Social Factors Influencing Repeated Pregnancy Among Unmarried Adolescent Mothers.

The study report number of social factors influencing repeated pregnancy among unmarried adolescent mothers. These include; ability to make decisions, peer pressure and parent-child relationship.

#### 3.2.5. Decision Making

Not being able to make sexual decisions was reported by key informants. This was characterized by inability to negotiate whether they are willing to have sex with their partners at a particular time or not, as well as the use of protection (condom). Some of the participants reported that they were forced to have sex by their partners. Moreover, some reported that men are sometimes not ready to use a condom. All of these led them to have repeated pregnancies. One of the key informants pointed out that:

“I didn’t like engaging myself in doing sex because I knew that I will be pregnant. Since he was forcing me and I was unable to resist because he was already my fiancée, lead me to become pregnant later. This is now the second pregnancy I am carrying”.(KII 02)

Another key informant also added that:

“No, the man is the one who was not ready to use the protection (condom). I was unable to deny him from that. So this made me get this pregnancy”(KII 15)

#### 3.2.6. Peer Pressure

Peer pressure was also revealed as one of the crucial factors influencing repeated pregnancy among unmarried adolescents. This is characterized by being lured, accused of being infertile, and fear of being gossiped about, i.e., that they can conceive only once. Some of the participants reported that adolescents sometimes sneak from their homes and go to watch cinemas; this is where they meet boys and, ultimately, end up having sexual intercourse with them. One of the key informants reported that:

“We are always told to remain at home. But at night we sneak from the houses and go to the cinemas. This environment makes it very easy to get pregnant. This is because of the types of houses that we have, adolescents have their houses built separately from their parents’ houses.”(KII 16)

Another participant reported that:

“I can’t stop conceiving because people will say that the child of a certain person is infertile or start asking what problem she has. It is very painful sometimes when they see you with one child they will say that she can conceive once.”(KII 14)

#### 3.2.7. Parent–Child Relationship

Poor relationships between adolescents and their parents emerged as one of the important factors influencing repeated pregnancy. This was characterized by the denial of the parents to take their children to family planning programs, not having time to discuss sexual-related issues with children, and sending children to do business, instead of going to school. One of the key informants pointed out that:

“No, I asked my mother to take me to the family planning program but she did not do so. Despite not knowing where the family planning centers are, my mother used to tell me to go there alone.”(KII 13)

#### 3.2.8. Another Key Informant Reported That

“When the parents send young children to engage in business, they start getting used to money too early. So when they lack, that is when they start asking themselves where to get them. This leads to engaging themselves in sexual relationship issues to get money.”(KII 12)

## 4. Discussion

The study reported that some unmarried adolescents had inadequate sexuality knowledge, since they never attended any sexual education-related seminar or program, and they lacked knowledge on family planning. The study found that some adolescent girls did not even have the knowledge of how to use condoms. Lacking education on sexuality led adolescents to repeat pregnancy. These findings are supported by the study performed in Nigeria, reviewing the problems of adolescent sexual behavior and role of Millennium Development Goals 4, 5, and 6, as it reported that one of the factors that exposed adolescents to unwanted pregnancy was the limited knowledge of safe sex [10]. The lack of comprehensive sexual education was also mentioned in a systematic review of determinants of adolescent pregnancy in sub-Saharan countries [4]. These findings imply the importance of sexual education among adolescents to prevent unwanted pregnancy.

In addition, participants reported the issue of fear of contraceptive side effects. It was found that some participants reported changes in their menstrual cycle and bleeding longer than usual as barriers to using the contraceptive. Not using contraceptives was one of the reasons for repeated pregnancy. The findings of the current study are supported by Yakubu and Salisu [4] in a systematic review, as they reported that the fear of contraceptive side effects seems to be a barrier to contraceptive use. A study performed in Iran also reported that women had inaccurate perceptions, regarding contraceptives (intrauterine devices), such as fear of pain, IUDs being larger than the genitalia, and sexual dysfunction [20]. While some of these are based on actual health-related side effects, many fears are based on rumors, rather than personal experience.

The results showed that adolescents had to conceive to keep their boyfriends because they have been promised that they are going to be married; however, these promises were never fulfilled. These findings are supported by a systematic review exploring factors that the shape young people’s sexual behavior, as it was reported that, in order to hold on to their boyfriends, adolescent girls agree to sex [21]. These findings imply that the fear of losing their boyfriends can be the reason for repeated pregnancy; however, since adolescents are sexually active, this poses them with a risk of engaging in sex, hence becoming vulnerable to falling pregnant.

It was found that men were the ones with the power to make decisions, regarding intimacy issues. When a man wants to have sex, women have no choice but to accept because they seem to have no right to negotiate. Some of the participants reported that they were even forced and did not use protection. These findings are supported by the study assessing socio-cultural influences in decision-making involving sexual behavior among adolescents in Khayelitsha, Cape Town, as it revealed that females were not able to negotiate for safe sexual practices [22]. The similarity between these findings may be because they are both from Africa, hence the similarity in some cultural issues. The findings imply that traditions privilege males and put females under males’ control, hence putting females at risk of getting unwanted/unintended pregnancies.

The parent–child relationship was posed as one of the important factors influencing repeated pregnancy among unmarried adolescents. Results established that some parents denied taking their children to attend family planning programs. Additionally, it was revealed that, instead of sending them to school, adolescents were sent to do business, which exposed them to more risk of having pregnancy repeatedly. The study exploring factors contributing to teenage pregnancy in the Capricorn district of the Limpopo Province supports the current study, as it revealed that repeated pregnancies may be associated with parents’ reluctance in making sex and contraceptive education available to their kids [23]. A study in the United States reported that teenagers with higher levels of parental guidance were less likely to engage in sexual intercourse [24]. The findings imply that parents may be interpreting providing sex and contraceptive education to their children as permission for them to engage in sexual activities.

## 5. Limitation

Since the study is qualitative and involves the purposive selection of the study sample, its findings cannot be generalized. The readiness of teenage mothers to provide the information regarding some of the reasons for becoming pregnant, since it is normally considered illegal and some explanations point directly to their relatives or teachers, they might have provided unreliable information. This was achieved by explaining to them that confidentiality is assured, no names were involved, and the information may be disclosed only for this research.

The study intended to interview adolescents aged 15–19 years. However, by default, only those aged 17–19 years participated in the interviews; hence, we might have missed important information from those aged below 17 years. However, to ensure that we obtained substantive information, probing was performed, and participants were encouraged to speak their minds.

## 6. Conclusions

The study identified several individual-level factors influencing repeated pregnancy among unmarried adolescents. These includes lack of sexuality education (for example, the proper knowledge of family planning methods), individually perceived barriers to contraceptive use (fear of side effects and not being educated on contraceptives), and the guarantee of marriage, as it was reported that some obtained a second pregnancy, since they were promised marriage. The study also reported the power to make decisions, peer pressure, and the parent–child relationship as social factors influencing repeated pregnancy among unmarried adolescents.

Unmarried adolescent mothers should abstain from having sex. Additionally, they should be able to say no to unwanted sex. The campaigns to raise awareness on sex education among adolescent girls should be emphasized by the Ministry of Health and other implementing partners dealing with reproductive health and should go along with those involving parents. Moreover, the Ministry of Health and other implementing partners dealing with reproductive health should strengthen campaigns to encourage gender balance by changing community attitudes towards the position/status of women in society as a whole.

Since this study did not include mothers of unmarried adolescents, we suggest further studies should be performed to obtain their perspective, regarding the factors influencing repeated pregnancy among unmarried adolescents.

## Figures and Tables

**Table 1 children-09-01523-t001:** Example of thematic analysis.

Verbatim	Code	Sub-Theme	Theme
Individual level factors influencing repeated pregnancy among unmarried adolescents			
“*I did not expect to hold pregnancy the second time. It was just bad luck because I did not plan for it. I did not know that I was in my dangerous days. I thought I would not be able to conceive in those days. Unfortunately, it happened* *“I do not know how condoms can be able to avoid pregnancy and by that time I even did not know in which ways it can prevent pregnancy. Although I was taught still I did not know how it worked*”.	Did not expect Bad luck Unaware of my dangerous days	Insufficient sexual education	Lack of sexual knowledge
	Condom use Did not know Menstrual cycle How condom works Lack of right information about family planning	Insufficient knowledge of family planning methods	

**Table 2 children-09-01523-t002:** Socio-demographic characteristics of the study participants.

Participants’ Reg #	Age	Marital Status	Education Level	Gravidity and Parity	N of Home Delivery	Reason for Home Delivery
KII01	19	Unmarried	STD 4	G2, P1, L1	None	N/A
KII02	19	Unmarried	STD 7	G2, P1, L1	None	N/A
KII03	19	Unmarried	STD 7	G3, P2, L2	None	N/A
KII04	19	Unmarried	STD 6	G3, P2, L2	None	N/A
KII05	18	Unmarried	STD 3	G2, P1, L1	None	N/A
KII06	19	Unmarried	STD 6	G2, P1, L1	None	N/A
KII07	18	Unmarried	STD 4	G2, P1, L1	None	N/A
KII08	18	Unmarried	STD 6	G2, P1, L1	1	Fastened labor
KII09	19	Unmarried	STD 4	G2, P1, L1	1	Ignorance
KII10	18	Unmarried	STD 7	G2, P1, L1	None	N/A
KII11	17	Unmarried	STD 5	G2, P1, L1	None	N/A
KII112	18	Unmarried	STD 2	G2, P1, L1	None	N/A
KII13	17	Unmarried	FORM 1	G3, P2, L2	None	N/A
KII14	19	Unmarried	STD 6	G2, P1, L1	None	N/A
KII15	19	Unmarried	Never went to school	G2, P1, L1	None	N/A
KII16	18	Unmarried	STD 7	G2, P1, L1	None	N/A

Key: G: GRAVIDA, P: PARA, L: LIVING, STD: standard.

**Table 3 children-09-01523-t003:** Emergent themes and sub-themes.

Theme	Sub-Theme
**Individual level factors**	
Inadequate sexuality knowledge	Insufficient sexual education
	Insufficient knowledge of family planning methods
Individual perceived barriers to contraceptive use	Fears of side effects among adolescents
	Inadequate knowledge of contraceptives knowledge
Guarantee for marriage	Fear of being dumped by their boyfriends
	Marriage obsession among adolescent girls
**Social level factors**	
Decision making	Inability to negotiate about sex
	Men are superior
Social pressure	Societal judgment
	Peer pressure
Parent–child relationship	Adolescents’ engagement in business
	Sexuality and family planning-related issues

## Data Availability

The authors are happy to share anonymized data related to this paper upon receiving a special request, along with the purpose of that request. Interested parties may contact mwajumammbaga49@gmail.com.

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
