# Peer review of "Individual and Social Level Factors Influencing Repeated Pregnancy among Unmarried Adolescent Mothers in Katavi Region—Tanzania: A Qualitative Study"

_children, 2022, doi:10.3390/children9101523_

Round 1

Reviewer 1 Report

Due to the small cohort of study no statistical conclusion can be reached.

Author Response

Response to Reviewer 1 Comments

Point 1: Due to the small cohort of study no statistical conclusion can be reached.

Response 1: This was a qualitative study, hence findings can not be generalized. However, with the number of participants interviewed we were able to reach saturation.

Reviewer 2 Report

children-1891118-peer-review-v1

This is a qualitative exploratory study to establish the individual and social factors influencing repeat pregnancy among single teenage mothers in a region of Tanzania.

The manuscript requires some methodological clarifications.

Notes to authors

Operational definitions

1.     Operational definitions require bibliographic support

Study context

1.     Why were adolescents under 15 years of age not included, if their operational definition includes them (10 to 19 years of age)? You only included adolescents between 17 and 19 years old.

Study design and sampling method

1.     What type was the qualitative approach: narrative or phenomenological?

2.     The research question needs to be clarified.

Data collection

1.     Why did only adolescents participate as key informants and not include mothers of adolescents?

2.     What was the interview guide?

Results

1.     The tables are not referenced in the text.

Author Response

Response to Reviewer 2 Comments

Operational definitions 1: Operational definitions require bibliographic support

Response 1: Bibliographic support has been provided to Operational definitions (Page 3).

Study context 2: Why were adolescents under 15 years of age not included, if their operational definition includes them (10 to 19 years of age)? You only included adolescents between 17 and 19 years old.

Response 2: This has been explained in the study limitations (Page 8). Adolescents under 15 years of age were not included because adolescent/teenage pregnancy is when a girl aged 15-19 is pregnant with her first child or gives birth. The study intended to interview adolescents aged 15 – 19 years, however, by default only those aged 17 – 19 years participated in the interviews, hence we might have missed important information from those aged below 17 years. However, to ensure that we get substantive information, probing was done and participants were encouraged to speak their minds.

Study design and sampling method 3:

  1. What type was the qualitative approach: narrative or phenomenological?
  2. The research question needs to be clarified.

Response 3:

  1. The qualitative approach adopted was a narrative (Page 3).
  2. The study intended to answer the following research question; What are the individual and social level factors that influence repeated pregnancy among unmar-ried adolescent mothers in Katavi Region? (Page 3).

Data collection 4:

  1. Why did only adolescents participate as key informants and not include mothers of adolescents?
  2. What was the interview guide?

Response 4:  

  1. The study intended to explore adolescents perspective, however, in the conclusion we have suggested further studies to obtain the perspective of mothers of adolescents regarding factors influencing repeated pregnancy among unmarried adolescents (Page 9)
  2. The interview guide was the key informant interview guide for unmarried adolescent mothers (Page 3)

Results 5: .     The tables are not referenced in the text.

Response 5: All tables have been references in the text (Page 4 – 5).

Reviewer 3 Report

Thank you for having opportunity to review this manuscript.This is a well-written one and shown some importment infirmation oa Africa countries health issues. I have  a concern:Page 4 ,"Education level",what does STD means?

Author Response

Response to Reviewer 3 Comments

Point 1: Thank you for having opportunity to review this manuscript.This is a well-written one and shown some importment infirmation oa Africa countries health issues. I have  a concern:Page 4 ,"Education level",what does STD means?.

Response 1: The STD means standard e.g. Standard 7 (Page 5). This has been described in the key under Table 2.

Round 2

Reviewer 1 Report

Accepted in the present form